# Association of Environmental Elements with Respondents’ Behaviors in Open Spaces Using the Direct Gradient Analysis Method: A Case Study of Jining, China

**DOI:** 10.3390/ijerph19148494

**Published:** 2022-07-12

**Authors:** Jing Zhao, Linshen Wang, Qing Ye, Qiang Zhao, Shutong Wei

**Affiliations:** 1School of Civil Engineering and Architecture, University of Jinan, Jinan 250022, China; cea_zhaojing@ujn.edu.cn (J.Z.); wst08012022@163.com (S.W.); 2School of Architecture and Art Design, Hebei University of Technology, Tianjin 300132, China; qingye@tju.edu.cn; 3Tianjin Bureau of Planning and Natural Resources, Tianjin 300042, China; bianshilan@126.com

**Keywords:** direct gradient analysis, open space, respondents’ behavior, environmental elements, GIS

## Abstract

Following rapid urban development, higher demands are now being placed on urban open spaces in China, and the relationship between environmental elements and respondents’ behaviors in open spaces has become a common concern for researchers. Current research using geographic information systems has yielded macroscopic portraits of the behavioral trends and outcomes of research subjects, but evaluating their actual needs is complex. This paper proposes a new method to analyze the relationship between open spaces and respondents’ behaviors from a detailed perspective. Direct gradient analysis was employed with stratified sampling to select sample points in open spaces. Environment quality, ancillary facilities, and canal culture were selected as subjective evaluation factors. The greatest advantage of the proposed procedure is that it produces a ranking diagram, which compensates for the shortcomings of research methods that cannot directly express the actual needs related to respondents’ behaviors. From a case study in Jining, China, a location’s environmental quality and ancillary facilities were found to have the greatest influence on the behaviors of those using open spaces. Finally, strategies for improving environmental quality in open spaces are proposed.

## 1. Introduction

In China, rapid urban development has had a significant impact on urban spaces and the behaviors of the users of the urban spaces. With the increasingly diversified and personalized lifestyle of the urban respondents, i.e., the users of open spaces (hereafter the users of open spaces are referred to as respondents), and their progressively broadening individual preferences and behaviors, higher demands have been placed on open spaces [1]. Humans are the creators and users of these spaces. In addition to the open spaces as objects of study during planning, design, and research, the needs and behaviors of people also need to be paid attention to. Considering the social attributes of open spaces, the behavior of respondents is regarded as an important aspect of open-space research [2,3]. In this context, the relationship between urban open spaces and respondents’ behaviors is a common concern for researchers [4]. At present, two broad types of methods are used: geographic information systems (GIS) and questionnaire surveys with quantitative analysis.

Research using GIS mainly focuses on the time–space behavior of respondents’ activities. For instance, Kwan et al. [5] applied geocomputation and geovisualization to discern human activity patterns; Leslie et al. [6] studied the associations of attractiveness, size, and proximity of multiple neighborhood open spaces with recreational walking; García-Palomares et al. [7] analyzed daily city dynamics in relation to land use; and Sletto and Palmer [8] combined GIS-based spatial analysis with rhythm analysis informed by phenomenological methods. However, the main problem with GIS analysis is that, although big data and GIS provide macroscopic portraits of research subjects’ behavioral trends and outcomes [9], it is difficult to evaluate the real needs or demands related to respondents’ behaviors or to discern the specific behavior of respondents to reach a certain destination. The real reason that respondents want to go to an open space—namely the elements of the open space that attract respondents to it—is obscure. Therefore, analyzing big data and applying GIS are not the most appropriate methods to accurately evaluate the relationship between specific small urban spaces and respondents’ behaviors.

The quantitative analysis of a questionnaire is considered more suitable for the study of small urban spaces in relation to the behaviors of a specific population. For example, Calderwood and Freathy [10] analyzed the behavior of respondents as consumers; Iamtrakul et al. [11] studied the behavior of respondents in relation to urban parks; and Huang et al. [12] researched thermal comfort and user behaviors in outdoor spaces. There is also growing empirical evidence of a positive relationship between the complex patterns of residential development and environmental quality in general [13]. Various authors have shown that environmentally integrated facilities in open spaces promote respondents’ well-being, such as, for example, by controlling urban landscapes both physically and symbolically [14] or by offering environmentally aware residential support [15]. However, these studies focus on some aspects of open space or respondents’ behaviors without considering the specific relationship between environmental elements of open space and respondents’ behaviors.

The significance of the strong relationship between the environmental elements of open spaces and the behavior of respondents is beyond doubt. The environmental elements of an open space include basic quality requirements for human settlements, ancillary facilities, and cultural features. It can help us determine exactly what elements attract respondents to open-space activities. In this way, designers can improve future construction or reconstruction planning to increase respondents’ use of open spaces. Therefore, to analyze the relationship between environmental elements of open space and respondents’ behaviors from a very detailed perspective and with a specific population as the research object, it was necessary to develop a new method.

This paper aimed to propose a new method for analyzing the relationship between open spaces and respondents’ behaviors from a detailed perspective. In the proposed procedure, direct gradient analysis (DGA) was used as the basic tool, and stratified sampling was used to select sample points in open spaces. Environmental quality, ancillary facilities, and canal culture were selected as subjective evaluation factors. Using Jining City (in eastern China) as an example, this paper extracts indices that significantly explain respondents’ behaviors through multivariate statistical analysis using respondents’ behaviors as response variables and environmental factors as explanatory variables. The types and features of environmental factors that affect respondents’ behaviors are elaborated in terms of the denotative meanings of these indices, and suggestions for improving open spaces in Jining are presented.

## 2. Materials and Methods

### 2.1. Study Area

To explore the relationship between respondents’ behavior and environmental factors using DGA, open spaces in Jining, an important city in eastern China, were selected (Figure 1). A 53.88 km^2^ built-up area of Jining, bordered by China National Highway 327 in the north, Jinghang Road in the south, Guangfu River in the east, and the Grand Canal in the west, was analyzed. It includes not only the inner and outer portions of Jining Old Town but also numerous water systems, such as the Grand Canal, Yue River, Guangfu River, and Wangmuge Lake. By using the above as the sampling area, not only is the overall situation of well-developed urban areas considered but all open spaces along and around the river and lake systems are incorporated.

### 2.2. Data Sources and Collection

Urban open spaces comprise numerous entities of varied and complex large-scale typologies. Before sampling, they should be classified into several categories and reclassified into subcategories based on differences in natural features, scale, function, and service scope. Stratified random sampling was used to determine the samples for this paper.

#### 2.2.1. Redefining “Open Space”

“Open space” is not yet recognized as a legal land-use category in China. There are many definitions of an “open space” [16]. Some researchers believe that an open space signifies an area that is outside of a building, outdoors, or not covered by a roof [17] or one that qualifies as a green-space system (other than a building lot) [18]. Some researchers understand open space from the medium perspective, which evaluates open space based on its urban cultural value and views it as a facilitator of activities across social groups [19]. Meanwhile, other researchers define open space using the feature perspective, which situates an open space as free to access, shared by respondents, and unenclosed [20]. Although there are different ways to define “open space”, the abovementioned approaches share a few commonalities: they all define “open space” as uncovered and used for urban activities. Therefore, an “open space” is an uncovered space in a built-up urban area. Accordingly, considerations of the open space in Jining should take into account the above two commonalities, as well as the waterfront space and the space around the unique historical and cultural relics in Jining City. In sum, the study of open space in Jining should not only include urban green space systems and commercial streets, but also spaces around riverbanks and historical and cultural sites. These open spaces in Jining correspond to the subjective behaviors of some respondents.

#### 2.2.2. Stratification

Open spaces were stratified into non-intersecting layers within the scope of this paper. According to the definition of open spaces provided herein, public green spaces, roadside green spaces, commercial streets, and attached green spaces within the built-up area were regarded as four layers for separate sample extraction.

#### 2.2.3. Sampling

The probability sampling method was used in combination with the results of the classification of open spaces. A stratified sampling approach was adopted, whereby at least two open spaces were randomly extracted from each layer. To cover the full range of open spaces in the Jining urban area, the random sampling points were evenly distributed across the area, and the linear distance of each sample point to its nearest sample point was not greater than 1000 m. A total of 12 sample points were selected, as shown in Figure 1. The sample points of public green spaces are People’s Park, Nanchi Park, Guanghe Park, and the Xianying green space. The sample points of the roadside green space are the Tieta Temple roadside green space, the Riverside green space at the east gate of the Yunhe Shengshi residential community, and the Kuaihuolin green space. The sample points of commercial streets are Wanda Plaza, Huanbiquan Road, and Yunhe Mall. The sample points of the attached green spaces are various activity venues and inter-dwelling green spaces in the Huijingyuan residential community and the Guiheyuan residential community.

#### 2.2.4. Data Collection

For the present survey, data were collected through questionnaires. The experiment was conducted on two days in July 2021 from 7:00 to 11:00 and 15:00 to 19:00 on each day. At least 30 respondents were selected randomly at each sample point for the interview, and the interviewers received oral forms of informed consent from them. To avoid excessively large score deviations caused by unfamiliarity with the sample points, most respondents selected during the interview were regular users of the studied open spaces. Interviews were conducted until the number of valid questionnaires reached 30 for each sample point, resulting in a total of 360 questionnaires being collected. This is because, in statistical research, the sample mean generally follows a normal distribution if the sample size exceeds 30, according to the central limit theorem [21].

#### 2.2.5. Identification of Evaluation Factors

Regarding respondents’ subjective assessments of urban open spaces, the evaluation factors in the questionnaire were designed in accordance with the following principles. First, the number of evaluation factors should be as small as possible to avoid prolonged data collection and a cumbersome analytical process. Nevertheless, they should reflect reasonably abundant information. Second, the evaluation factors should not be excessively detailed while still covering the problems under study to avoid interference with the overall analysis results. Third, the evaluation factors should cover basic but adequately inclusive information regarding the correlation between respondents’ behaviors and environmental variables.

Based on the above principles and the four basic quality requirements for human settlements (safety, health, convenience, and comfort) as stipulated by the World Health Organization [22], safety, cleanliness, beauty, and convenience were designated as evaluation factors for the first major category, environmental quality. According to the infrastructure conditions around the survey sites, leisure, catering, entertainment, and tourist facilities were designated as evaluation factors for the second major category, ancillary facilities [23]. Notably, respondents are not only demanding more today from the physical amenities of urban spaces, but also from their cultural aspects [24]. Historically, Jining was the residence of the highest management organization of the Grand Canal, and the city is full of relics of the Grand Canal and its management organization. The people of Jining attach great importance to whether they can perceive the Grand Canal in an open space. Perception refers to vision and atmosphere. Visual perception specifically refers to the landscape including the Grand Canal, and atmosphere perception refers to whether the open space has Grand Canal-related folk culture and artistic forms. In the questionnaire interview, some respondents suggested that the activities, exhibitions, or sketches in the open space, there to help people understand the history and culture of the Grand Canal, were among the reasons that attracted them. Accordingly, respondent perceptions of canal culture (e.g., canal landscape conservation and the cultural atmosphere of the Grand Canal area) was set as the third category.

All ten factors or subcategories under the three major categories (Table 1) were evaluated on an equidistant five-point Likert-type summated rating scale. Each variable under the categories of environmental quality and ancillary facilities was rated on a scale ranging from very unsatisfied (1 point) to very satisfied (5 points). For the variable on the canal’s cultural atmosphere, an alternative hierarchical scale was designed, which was similarly scored 1–5 points according to the respondents’ perceptions.

#### 2.2.6. Direct Gradient Analysis of Correlation between Open Spaces and Respondents’ Behaviors in Jining

To understand the role of open spaces in respondents’ daily activities in Jining, the behaviors of respondents at the time of the survey were also recorded in the questionnaires. This paper only focused on their leisure behaviors in open spaces. “Leisure” refers to non-forced and internally motivated activities that involve free time and are not caught up in work, self-management, or sleep [25]. Combined with relevant research [26,27] and field observations, the behaviors in Jining’s public spaces were grouped into seven categories: physical exercise, cultural activities, meeting friends, parent–child activities, sightseeing, passing through, and other. There are many kinds of leisure activities. The first six kinds of leisure activities listed here are common in Jining’s open spaces; uncommon leisure activities fall under the seventh category.

To achieve precise analysis by associating respondents’ behaviors and environmental factor variables with the corresponding sample points for case-by-case analysis, DGA was used to conduct a multivariate statistical analysis of the correlation between those behaviors and factors. The data consisted of two parts: response variables (behavioral composition) and explanatory variables (environmental factors). DGA can explore how explanatory variables affect response variables and, therefore, how open-space environmental factors impact the distribution of respondent behaviors.

Canoco 5 for Windows was used to determine the correlation between behavior types and open-space environmental factors. This program can intuitively express the positive and negative relationships between behavior types and open-space environmental factors using two-dimensional sorting diagrams [28]. This supplements other correlation analyses and therefore enables relatively more comprehensive and clear conclusions.

### 2.3. Analysis Models

Association of environmental elements with respondents’ behaviors in open spaces is a complex research topic [29]. Many important influencing factors, such as the environment, facilities, and culture, work on open spaces simultaneously [30,31,32]. The diversity of respondents’ behavior in open spaces has also been difficult to address [33,34].

It is noteworthy that the elements composing open spaces and the behaviors of individuals in those spaces are two different sets of multivariate data. As a commonly used analysis method in ecology [35], the technique of DGA is well-developed and can be used to analyze the correlation between two multivariate datasets to obtain credible results [36], such as to display the distribution of organisms along gradients of important environmental factors. The basic idea of this paper was to apply DGA to the analysis of urban open space because it is an analysis method for two groups of variables.

DGA primarily includes canonical correspondence analysis (CCA) and redundancy analysis (RDA). While CCA is based on a unimodal model, RDA is based on a linear model. In the unimodal model, the number of certain respondents’ behaviors increases as the value of a certain environmental factor increases. When the environmental factor increases to a certain value, the number of individuals with the behavior reaches a maximum; the value of the environmental factor at this time is called the optimum of the behavior. When the environmental factor value continues to increase, the number of behaviors gradually decreases. In the linear model, a respondent’s behavior changes linearly with the change in a certain environmental factor. Whether to use CCA or RDA in DGA should be determined on the basis of a detrended correspondence analysis (DCA) of the response variables to ascertain the type of relationship model. DCA, an ordination technique, represents assemblage samples as points in multi-dimensional space; similar assemblages are located close together and dissimilar assemblages farther apart [37]. After analyzing the response variable using DCA, the gradient length of the four axes will be obtained. If the gradient length for the first axis is greater than 4.0, CCA should be used, but if the value is in the range of 3.0–4.0, both CCA and RDA can be used, and if it is less than 3.0, RDA yields better results than CCA. Generally, CCA is preferred for performing DGA, but when CCA results in poor ordination, RDA may be preferable [38].

## 3. Results

A DCA was initially conducted using response variable data to examine whether the relationship between response and explanatory variable data satisfied the conditions of RDA. The DCA results revealed that the gradient length of the first axis was 1.865. As the gradient length of the first axis was less than 3.0, which is a condition wherein RDA yields better results than CCA, for Jining’s open spaces, RDA was used to ordinate respondents’ behaviors. From the classifications based on the survey of respondents’ behaviors, the behavioral types at each sample point were obtained as percentages (Table 2).

### 3.1. Demographic Information on Respondents

Among the 360 questionnaire respondents, 328 (91%) were local respondents, thus satisfying the requirement that most respondents be familiar with the sample points. The male-to-female ratio was 1:0.85. The age ratio of young (18–40) to middle-aged (41–59) to elderly (60 and older) [39] respondents was 1:0.92:0.63. In the present survey, age groups were chosen according to the All-China Youth Federation and the normal retirement age in China. Regarding educational level, 58% of the respondents had senior high school education or below, whereas 42% of the respondents had a junior college degree or above. Nearly two-thirds of the respondents had a monthly income lower than RMB 3500, whereas less than one-third had a monthly income of RMB 3500–6000. Regarding occupational distribution, retirees constituted the largest number of open-space users, followed by freelancers and company employees (Figure 2). Students, workers, and homemakers were active users of open spaces. Environmental quality was the most important factor affecting the respondents’ behavior.

The obtained RDA ordination diagram of Jining respondents’ behaviors and environmental factors is shown in Figure 3, in which black dots represent the 12 Jining urban sample points, and arrows represent the respondents’ behaviors and environmental factors. The black radiating dashed arrows indicate respondents’ main behaviors in Jining’s open spaces, and the red radiating solid arrows indicate the evaluation factors. Longer arrows imply a greater influence of environmental factors. The angles between the arrows reflect the correlations among environmental assessment factors: an acute angle indicates a positive correlation, and an obtuse angle indicates a negative correlation. Accordingly, leisure facilities, cleanliness, tourist facilities, safety, and beauty are negatively correlated with entertainment facilities, historical features, and cultural atmosphere. Exercise behavior is highly positively correlated with leisure facilities and a clean environment. Meeting friends and cultural activities, on the other hand, are strongly positively correlated with catering and entertainment facilities, as well as with convenience.

The RDA ordination diagram not only explains the relationship between respondents’ behaviors and environmental factors but also reflects the similarities and differences among various behaviors regarding their environmental requirements. For example, meeting friends, cultural activities, and other (unspecified) behaviors place fundamentally identical demands on environmental factors; respondents engaging in these behaviors are inclined toward places with good catering facilities and convenient transportation. Parent–child activities and sightseeing have consistent environmental requirements, with respondents who participate in these behaviors preferring environmentally safe places with beautiful scenery and convenient transportation. In contrast, the physical exercise group differs significantly from other behavioral groups by exhibiting a propensity for clean and neat environments with good leisure facilities.

From Figure 3, one can see that the closer a behavior plots to a sample point, the higher its frequency of occurrence. Similarly, the distances of the two behaviors from a sample point can also represent the difference in their proportions at that site. It is clear that exercise behavior occurs most frequently at Sample Point 6 (Guanghe Park), followed by Sample Points 7 (Xianying green space), 10 (Kuaihuolin riverside green space), 2 (Yunhe Mall), and 4 (People’s Park). According to the respondents, these sites share the features of abundant leisure facilities and a clean, neat environment (Table 3). Sample Point 5 (Nanchi Park) is the site with the most frequent parent–child activities and sightseeing behaviors. After incorporating environmental factors, Nanchi Park was found to have the most convenient transportation and the most beautiful environment. Cultural activities and gatherings of friends take place chiefly at Sample Points 12 (Guiheyuan), 8 (Tieta Temple), and 1 (Wanda Plaza), indicating that people prefer places with abundant catering facilities and convenient transportation to meet friends and for cultural activities, such as art performances.

### 3.2. Verification

#### 3.2.1. Validating Research Method

From the above survey methods, the degree of satisfaction with various factors at the 12 sample points of Jining’s urban open spaces was calculated (Table 3). Subjective factors related to urban spaces in Jining, such as transportation accessibility and features of interest, were analyzed using point-of-interest (POI) analysis and GIS to validate the RDA analysis data and confirm the scientific and rational application of respondents’ open-space behaviors.

#### 3.2.2. Walking Accessibility

To allow quantitative calculation, walking accessibility was selected as the environmental quality to verify the reliability of the field survey data. In ArcGIS, the Network Analyst module was used to obtain the accessibility analysis diagram (based on 5-, 10-, and 15-minute time periods) and then to calculate accessibility areas for the 12 sample points (based on the same time periods; Figure 4).

Although the walking accessibility rankings for some sample points are different from those based on the questionnaire, the correlation between convenience satisfaction and the 10 min time period was 0.816 and was significant at the 0.01 level (two-tailed), which indicates that the questionnaire results are consistent with the actual situation (Table 4).

#### 3.2.3. POI Kernel-Density Analysis

To verify the ancillary facilities category, POI data for catering, entertainment, and tourist facilities were extracted for quantitative analysis, and then ArcGIS Spatial Analyst was employed for kernel-density analysis (Figure 5, Table 2). Correlation analysis of specific figures obtained for catering, entertainment, and tourism facilities yielded coefficients of 0.650, 0.656, and 0.681, respectively (Table 5). When these three indicators were used for verification, the results of the questionnaire survey were consistent with the POI kernel-density analysis and GIS results (Figure 5), indicating that the survey results reflect the actual situation.

## 4. Discussion

### 4.1. Characteristics of the DGA

Previous studies have emphasized the importance of the relationship between the environmental elements of open spaces and the behavior of respondents [40,41,42,43]. Using GIS and the quantitative analysis of a questionnaire, many studies have been carried out on elements of the open space that attract respondents [44,45,46]. This paper explored a new way of examining the influence of environmental elements on respondents’ behaviors by using DGA.

There are unique advantages to using this approach to explore the relationship between environmental factors and user behavior in open spaces. Verification experiments showed that the results of this method are consistent with the GIS method. However, DGA can more clearly and graphically reflect the relationship between different environmental factors and user behavior in multiple open spaces. The advantage of the DGA method is that to some extent it can compensate for GIS’s failure to provide details.

### 4.2. Discussion on the Environmental Elements and Respondents’ Behaviors in Open Spaces

Many environmental elements influence respondents’ behaviors, but the mechanisms remain unclear. Selecting among different open spaces with different characteristics is important because different open spaces complement one another [47]. When asked about the environmental elements that they liked about the open spaces, the respondents mentioned trails, shade trees, activity grounds, and river views. This scenario is consistent with the correlation analysis of this paper. Kaczynski et al. [48] found that park facilities, including paved trails, water area, and playgrounds, are more important than park amenities, such as drinking fountains, picnic areas, and restrooms for physical activity. This paper also found a significant relationship between the use of open spaces and leisure facilities, such as public chairs, pavilions, and tree shade; no significant relationship was found between the use of open space amenities, such as chess and card rooms, swimming pools, and fitness facilities. Petrunoff et al. [49] found respondents’ behaviors associations between facilities for active and passive recreation including children’s playgrounds, adult fitness corners, and hard courts for ball games. These findings are generally consistent with those of this paper except for a few inconsistencies. These inconsistencies could be because different types of open spaces have distinct characteristics of influences on use patterns.

## 5. Conclusions

This paper aims to explain the relationship between environmental elements and the behavior of respondents using the sequence diagram produced by the proposed procedure. It is found that the ranking chart helps to compensate for the shortcomings of other research methods that cannot directly express the actual needs or needs related to the behavior of the interviewees. At the beginning of the analysis, response variable data DCA is used to confirm whether the relationship between the response and the explanatory variable data satisfied the RDA conditions. Then it is concluded that, for the example in this article, RDA was the preferred method of ranking the behavior of the Jining respondents. In addition, stratified random sampling was used to determine specific samples from the Jining study area. According to the central limit theorem, it was determined that the number of samples collected in the data met the requirement of normal distribution in statistics.

The user behavior in Jining’s open spaces is not closely related to the layering and order of sampling points and the environmental components of sampling points in open spaces have the greatest impact on respondents’ subjective behaviors and related needs. Open spaces with abundant leisure facilities, especially those with public chairs, tree shade, fitness equipment, and other facilities, are more attractive than other open spaces. The reliability of the survey data was verified using POI and GIS analysis.

Through DGA, the correlation between environmental factors and respondents’ behaviors in open spaces can be objectively measured and used in other cities to assess whether the city’s open-space environment is suitable for certain activities. For environmental planners and urban designers, this paper provides possible improved targeted design strategies that can be combined with analysis results in future reconstruction plans or buildings to increase people’s use of open spaces. At the same time, the results obtained using this research method are conducive to improving the environmental quality of open spaces in other urban areas and can provide a solid foundation for the revitalization and transformation of open spaces worldwide.

It should be noted that the sample size in this paper is limited to the geographic area. Future research should focus on comparing the impact of the environmental element characteristics of different types of open spaces on the utilization of open spaces of different groups. Comparative studies could also be conducted after performing additional case studies.

## Figures and Tables

**Figure 1 ijerph-19-08494-f001:**
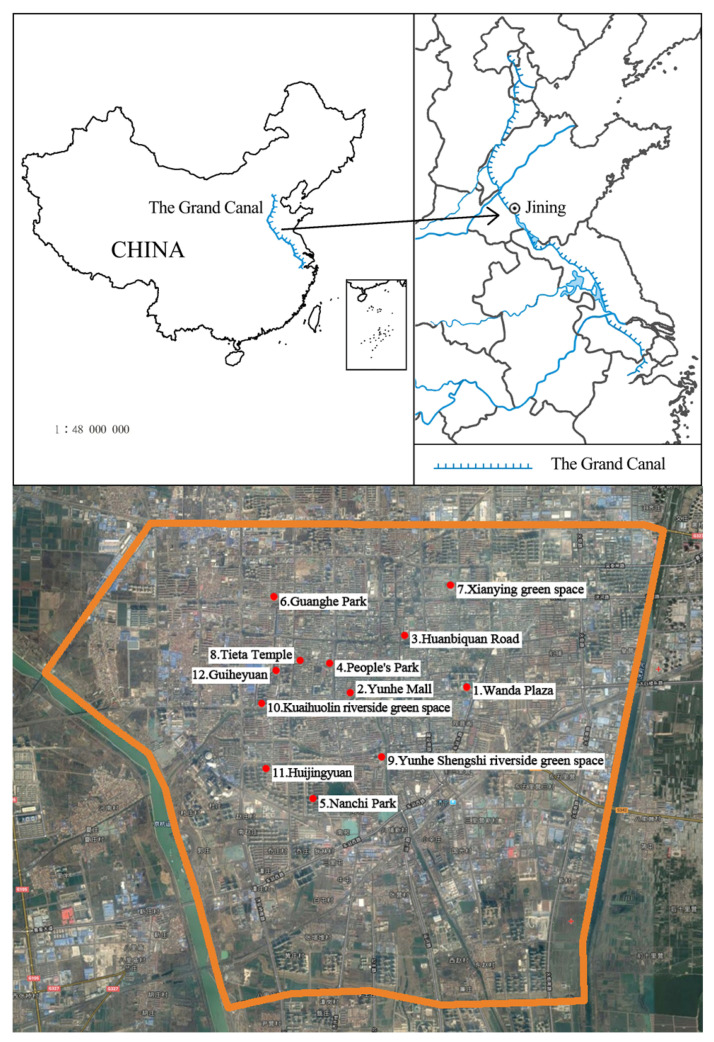
Study area and sample point distribution map for Jining urban space perception survey.

**Figure 2 ijerph-19-08494-f002:**
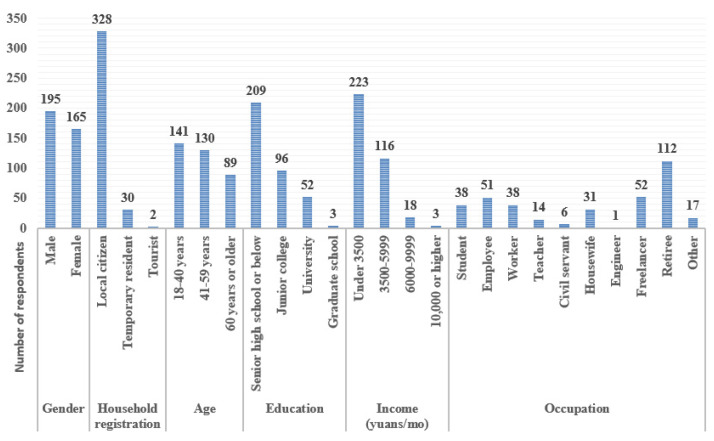
Basic information on respondents to Jining open-space questionnaire survey.

**Figure 3 ijerph-19-08494-f003:**
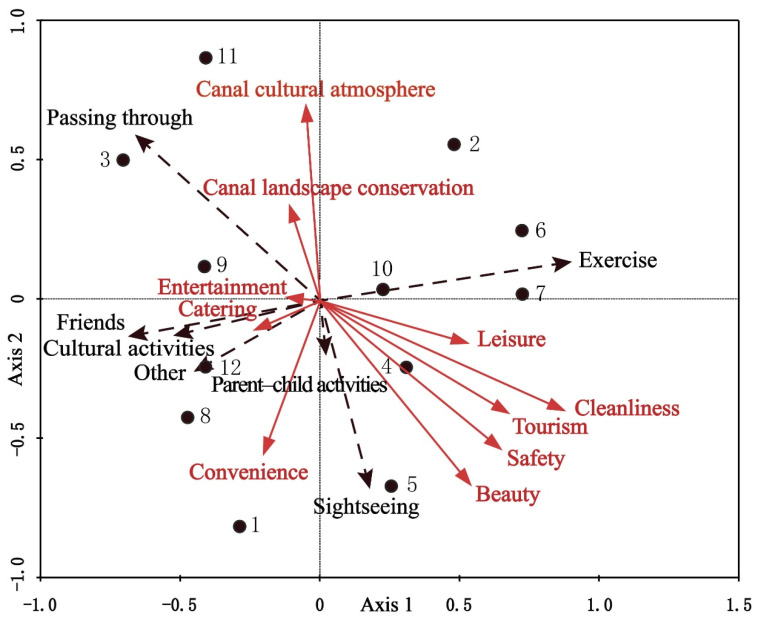
Redundancy analysis (RDA) ordination diagram of Jining respondents’ behaviors and environmental factors at 12 sample points.

**Figure 4 ijerph-19-08494-f004:**
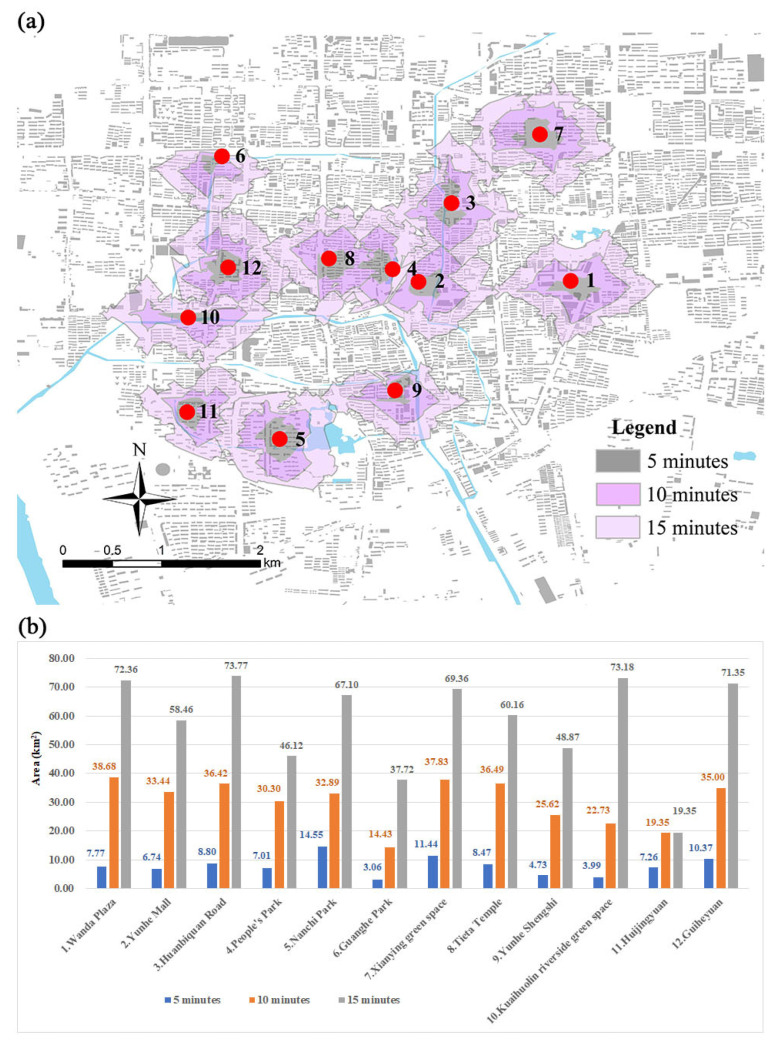
(**a**) Walking accessibility analysis of 12 sample points. (**b**) Comparison of areas of 12 sample points according to 5-, 10-, and 15-minute walking accessibility.

**Figure 5 ijerph-19-08494-f005:**
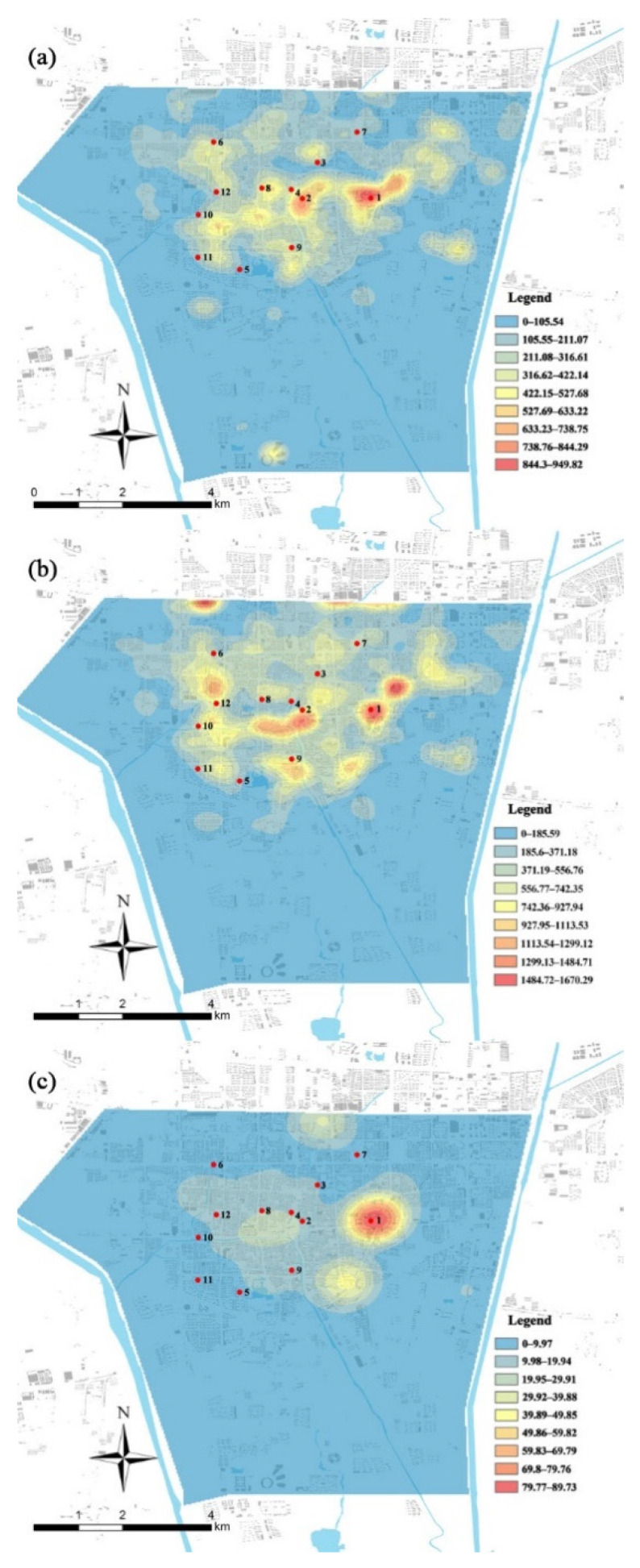
POI analysis of catering, entertainment, and tourism facilities: (**a**) POI analysis of catering facilities; (**b**) POI analysis of entertainment facilities; (**c**) POI analysis of tourism facilities.

**Table 1 ijerph-19-08494-t001:** Classification and definition of subjective factors evaluating urban spaces in Jining.

Major Category	Subcategory	Definition	Rating Description
Environmental quality	Safety	Security situation and disaster-prevention facilities.	Very unsatisfied = 1 point Unsatisfied = 2 points Fair = 3 points Satisfied = 4 points Very satisfied = 5 points
	Cleanliness	Clean and without pollution or noise.	
	Beauty	Buildings and landscapes are pleasing to the eye.	
	Convenience	Places are easily accessible.	
Ancillary facilities	Leisure facilities	Public chairs, pavilions, fitness equipment, tree shades, drinking-water systems, etc.	
	Catering facilities	Convenience stores, restaurants, pubs, etc.	
	Entertainment facilities	Chess and card rooms, ballrooms, karaoke televisions, swimming pools, etc.	
	Tourist facilities	Hotels, health resorts, parking lots, etc.	
Canal culture	Canal landscape conservation	Scenic features of the canal are preserved.	
	Canal cultural atmosphere	Jining’s unique canal cultural connotations, including folk culture and artistic forms.	No perception at all = 1 point Not strong = 2 points Moderate = 3 points Strong = 4 points Very strong = 5 points

**Table 2 ijerph-19-08494-t002:** Types of behaviors and specific values of point-of-interest (POI) kernel-density analyses of 12 sample points in Jining’s open spaces.

Sample Point		Types of Behaviors at 12 Sample Points (%)							Specific Values of POI Kernel-Density Analyses of 12 Sample Points (per km^2^)		
		Physical Exercise	Cultural Activities	Meeting Friends	Parent–Child Activities	Sightseeing	Passing through	Other	Catering Facilities	Entertainment Facilities	Tourism Facilities
1	Wanda Plaza	13.33	6.67	20.00	13.33	23.33	6.67	16.67	889.44	87.24	1550.40
2	Yunhe Mall	63.33	6.67	10.00	6.67	0.00	13.33	0.00	742.76	18.56	863.69
3	Huanbiquan Road	3.33	10.00	20.00	13.33	0.00	40.00	13.33	137.15	10.10	433.94
4	People’s Park	44.83	6.90	13.79	13.79	10.34	0.00	10.34	409.18	20.90	296.05
5	Nanchi Park	36.67	0.00	0.00	33.33	23.33	0.00	6.67	78.79	5.39	243.42
6	Guanghe Park	66.67	0.00	0.00	26.67	0.00	0.00	6.67	314.69	6.02	469.85
7	Xianying green space	70.97	0.00	0.00	9.68	16.13	3.23	0.00	159.61	7.80	409.64
8	Tieta Temple	6.67	10.00	23.33	20.00	0.00	10.00	30.00	388.62	19.86	405.27
9	Yunhe Shengshi	13.33	6.67	26.67	20.00	3.33	23.33	6.67	321.26	12.84	809.70
10	Kuaihuolin riverside green space	43.33	6.67	0.00	10.00	0.00	6.67	33.33	173.97	6.06	560.37
11	Huijingyuan	16.67	0.00	6.67	20.00	0.00	43.33	13.33	141.63	3.24	277.69
12	Guiheyuan	16.67	6.67	13.33	10.00	6.67	20.00	26.67	209.96	13.89	685.02

**Table 3 ijerph-19-08494-t003:** Degree of satisfaction with various factors at 12 sample points of urban open spaces in Jining (%).

Sample Point	Safety	Cleanliness	Beauty	Convenience	Leisure Facilities	Catering Facilities	Entertainment Facilities	Tourism Facilities	Canal Landscape Conservation	Canal Cultural Atmosphere
Wanda Plaza	90.00	96.67	100.00	100.00	80.00	96.67	93.33	76.67	96.67	86.67
Yunhe Mall	93.33	93.33	83.33	86.67	93.33	90.00	46.67	56.67	100.00	76.67
Huanbiquan Road	70.00	80.00	73.33	90.00	66.67	46.67	33.33	36.67	83.33	86.67
People’s Park	93.10	100.00	100.00	89.66	86.21	79.31	41.38	58.62	89.66	96.55
Nanchi Park	93.33	96.67	96.67	86.67	63.33	16.67	36.67	36.67	56.67	46.67
Guanghe Park	96.67	100.00	80.00	73.33	60.00	46.67	56.67	16.67	96.67	50.00
Xianying green space	100.00	96.77	90.32	93.33	38.71	3.23	3.23	19.35	93.55	93.55
Tieta Temple	100.00	83.33	70.00	100.00	30.00	0.00	0.00	20.00	83.33	100.00
Yunhe Shengshi	96.67	93.33	83.33	80.00	90.00	73.33	60.00	33.33	83.33	66.67
Kuaihuolin riverside green space	90.00	96.67	86.67	70.00	56.67	20.00	13.33	30.00	83.33	73.33
Huijingyuan	83.33	90.00	80.00	73.33	80.00	70.00	33.33	20.00	43.33	70.00
Guiheyuan	93.33	96.67	80.00	76.67	63.33	46.67	33.33	36.67	63.33	63.33

Notes: The degree of satisfaction was calculated as the percentage of respondents (out of the total respondents at the sample point) who rated the factor at 4 points or higher on a five-point Likert satisfaction-rating scale, where 1 represented “very unsatisfied” and 5 represented “very satisfied”. Thus, respondents’ behaviors were not closely linked to the stratification or hierarchy of sample points. The environmental composition of the sample points—in other words, a location’s environmental quality and ancillary facilities—was the most influential factor affecting respondents’ behaviors.

**Table 4 ijerph-19-08494-t004:** Correlation between convenience satisfaction and 5-, 10-, and 15-minute walking-accessible areas.

		Convenience Satisfaction	5 min.	10 min.	15 min.
Convenience satisfaction	Pearson Correlation	1	0.446	0.816 **	0.402
	Significance (two-tailed)		0.146	0.001	0.195
		*n* = 12	*n* = 12	*n* = 12	*n* = 12
5 min.	Pearson Correlation	0.446	1	0.666 *	0.411
	Significance (two-tailed)	0.146		0.018	0.185
		*n* = 12	*n* = 12	*n* = 12	*n* = 12
10 min.	Pearson Correlation	0.816 **	0.666 *	1	0.722 **
	Significance (two-tailed)	0.001	0.018		0.008
		*n* = 12	*n* = 12	*n* = 12	*n* = 12
15 min.	Pearson Correlation	0.402	0.411	0.722 **	1
	Significance (two-tailed)	0.195	0.185	0.008	
		*n* = 12	*n* = 12	*n* = 12	12

** Correlation is significant at the 0.01 level (two-tailed). ***** Correlation is significant at the 0.05 level (two-tailed).

**Table 5 ijerph-19-08494-t005:** Correlation between facilities kernel density and facilities satisfaction.

Catering Facilities		Kernel Density	Satisfaction
Kernel density	Pearson Correlation	1	0.650 *
	Significance (two-tailed)		0.022
	N	12	12
Satisfaction	Pearson Correlation	0.650 *	1
	Significance (two-tailed)	0.022	
	N	12	12
**Entertainment facilities**		**Kernel density**	**Satisfaction**
Kernel density	Pearson Correlation	1	0.656 *
	Significance (two-tailed)		0.020
	N	12	12
Satisfaction	Pearson Correlation	0.656 *	1
	Significance (two-tailed)	0.020	
	N	12	12
**Tourism facilities**		**Kernel density**	**Satisfaction**
Kernel density	Pearson Correlation	1	0.681 *
	Significance (two-tailed)		0.015
	N	12	12
Satisfaction	Pearson Correlation	0.681 *	1
	Significance (two-tailed)	0.015	
	N	12	12

* Correlation is significant at the 0.05 level (two-tailed).

## Data Availability

All data generated or analyzed during this study are included in this published article.

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
