# Peer review of "Association of Environmental Elements with Respondents’ Behaviors in Open Spaces Using the Direct Gradient Analysis Method: A Case Study of Jining, China"

_ijerph, 2022, doi:10.3390/ijerph19148494_

Round 1
Reviewer 1 Report
In general, the theme is on average interesting.
First of all, it is important to check the meaning of "Respondents' Behaviors" in Open Spaces: is the meaning "the behavior of users in open spaces"? or the meaning is the behavior of the users interviewed?
It is not clear how the methodology applied to a single case could also have value on others, with surveys carried out in a few hours of two days of the same month. Perhaps the users of the open spaces at other times or on other days of the week, or in another season could have provided different data.
Paragraph 6, compared to the methodology illustrated in the article, is of little help. The conclusions could be expanded.
Please, note that the images in figure 5 are too small and are not clear.
Reviewer 2 Report
The paper presents new method to analyze the relationship between open spaces and respondents’ behaviors from a detailed perspective.
The structure of the paper is adequate organized and tables and figure are informative. The paper demonstrates and cites, to some point, appropriate range of literature sources.
I have to point out that I am not commenting methods used as it not my key expertise. Nevertheless, I would like to state that presented research is appropriate designed and applied methods adequately described.
I have to stress that I assigned major revision as I am of the opinion that main limitations within presentation of this topical and comprehensive research/study is that some of the important information/data has/have not been presented/explained/defined.
To the authors is suggested following:
- Abstract ///line 19: “this study”…consider to replace into “this paper” proposes; line 78 – “This study aimed to propose” ... you meant the study/research performed? consider to correct the use of term “study” and stay consistent throughout whole manuscript;
- It will be of significance to list what it is consider under term environmental elements first mentioned;
- Consider to avoid using personal pronouns “we” (line 38, 73, 409..), “our” (line 407) etc.; please correct throughout whole paper;
- Line 36… Humans are creators and users of these spaces [2]..is this quote?
- Line 92 “Nicol and 90 Blake [17] noted that there is no clear concept of open space in normative or legal terms globally, except within academic literature, in which prior researchers have defined open space from three perspectives.”- this reference is from 2000.; please consider to update;
- Line 99 - “After comprehensively considering these three viewpoints, the open spaces evaluated in this study are defined as spaces that are uncovered by buildings and within the urban built-up area.” - what do you consider under “comprehensively considering these three viewpoints”-please explain; Line 184: “After consulting relevant research for methods of classifying respondents’ leisure activities [25], the behaviors in Jining’s public spaces were grouped into seven categories: physical exercise, cultural activities, meeting friends, parent-child activities, sightseeing, passing through, and other.”- what do you consider under “After consulting…” – please explain;
- Lines 195-198 – for data/information presented could you refer to the source/s?
- What is case study? Grand Canal region of China or Jining, China; not clear; please correct;
- Please provide references throughout whole manuscript (Line 244 – “All-China Youth Federation and the normal retirement age in China.”, etc.);
- I would like to propose to merge Section 5 and 6 into one (part 6 is related to the practice / future prospects/identifies certain implications within current practice and thus important part of the paper): Conclusions;
The authors are suggested to revise the paper carefully as some of the aspects of the manuscript need to be improved in order to support publication. Please refer to the Journal guidelines for authors.
Round 2
Reviewer 2 Report
I would like to stress that authors have had put efforts to correct paper and that important information/data has/have been adequately presented/explained/defined.
Further, I have to notice that the structure of the paper is, to some point, adequate organized and tables and figures are informative. The paper demonstrates and cites appropriate range of literature sources.
I have to point out that I am not commenting methods used as it not my key expertise. Nevertheless, I would like to state that presented research is appropriate designed and applied methods adequately described
However, I do still have some recommendations as some of the aspects of the manuscript need to be improved in order to support publication. To the authors is suggested following:
- - Abstract: consider to use other word/term for difficult, maybe complex, demanding…; (line 19):
- - It is suggested to explain “respondents” first time mentioned within line 33 (it is explained within line 43);
- -As stated previously consider to avoid using personal pronouns “our” (line 403, 409..); please correct throughout whole paper;
- - Please make clear when referring throughout whole manuscript: this paper presents the study/research performed, the study results instead of this study etc.; as suggested consider to correct the use of term “study” and stay consistent throughout whole manuscript (line 243…the basic idea of the study performed not this study….);
- - Line 130 - “..provided herein…” ; what do you mean? Where?
- - Line 181; what do you consider under term canal atmosphere? “ancillary facilities” -please provide reference;
- -Line 234 – “Open space is a complex research topic [23]”..is this quote?
- - The authors have had corrected Conclusions as suggested (merge Section 5 and 6 into one); however Conclusions still need further correction; it is suggested not to use subtitles within Conclusions; as the Conclusions should tie together to the other elements of paper, I would like to remind the author/s that a Conclusion should be, if possible, short; further, it should include summarized paper’s main points and does not contain any new information; also, it is advised to provide future prospects within Conclusions, all together with providing explanation of the value of the manuscript for similar research in related fields.
To the authors are suggested to revise the paper carefully as some of the aspects of the manuscript need to be improved in order to support publication. Please refer to the Journal guidelines for authors.
